# Case Study: Improving the Quality of Dairy Cow Reconstruction with a Deep Learning-Based Framework

**DOI:** 10.3390/s22239325

**Published:** 2022-11-30

**Authors:** Changgwon Dang, Taejeong Choi, Seungsoo Lee, Soohyun Lee, Mahboob Alam, Sangmin Lee, Seungkyu Han, Duy Tang Hoang, Jaegu Lee, Duc Toan Nguyen

**Affiliations:** 1National Institute of Animal Science, Rural Development Administration, Cheonan 31000, Chungcheongnam-do, Republic of Korea; 2ZOOTOS Co., Ltd., R&D Center, Anyang 14118, Gyeonggi-do, Republic of Korea

**Keywords:** dairy cow, point cloud registration, machine learning, deep learning

## Abstract

Three-dimensional point cloud generation systems from scanning data of a moving camera provide extra information about an object in addition to color. They give access to various prospective study fields for researchers. With applications in animal husbandry, we can analyze the characteristics of the body parts of a dairy cow to improve its fertility and milk production efficiency. However, in the depth image generation from stereo data, previous solutions using traditional stereo matching algorithms have several drawbacks, such as poor-quality depth images and missing information in overexposed regions. Additionally, the use of one camera to reconstruct a comprehensive 3D point cloud of the dairy cow has several challenges. One of these issues is point cloud misalignment when combining two adjacent point clouds with the small overlapping area between them. In addition, another drawback is the difficulty of point cloud generation from objects which have little motion. Therefore, we proposed an integrated system using two cameras to overcome the above disadvantages. Specifically, our framework includes two main parts: data recording part applies state-of-the-art convolutional neural networks to improve the depth image quality, and dairy cow 3D reconstruction part utilizes the simultaneous localization and calibration framework in order to reduce drift and provide a better-quality reconstruction. The experimental results showed that our approach improved the quality of the generated point cloud to some extent. This work provides the input data for dairy cow characteristics analysis with a deep learning approach.

## 1. Introduction

Unlike 2D photos, a 3D point cloud contains more information about an object or a particular environment. Specifically, the distance between objects could be measured by a 3D point cloud. Therefore, 3D point clouds are constructed and processed to provide a description of 3D geometry of objects. However, it is rather challenging to capture all the point cloud data of an object in one scan. Thus, we commonly move the 3D receiver around the object to obtain various point cloud at different viewpoint for the 3D shape of object reconstruction. These point clouds are registered to generate a comprehensive point cloud of the scanned object. The matching two-point cloud obtained at two different coordinates is called point cloud registration. For dairy cows, reproductive efficiency, milk quality, as well as farm life and productivity are determined by physical examination and linear testing of the cow. Specifically, each body component structure of a dairy cow affects its reproductive function and milk production efficiency. Therefore, the reconstruction of the dairy cow body for weight measurement and fitness computation has to be focused on and investigated extensively. From the point cloud of dairy cow, we could evaluate and give corresponding evaluation criteria based on the influence of each portion on its physiological quality via current approaches such as using neural network [1] instead of manual measurement and computation.

Currently, there are several techniques to create a 3D point cloud of objects. However, for moving objects such as animals or too large objects like tall buildings, generating a full 3D point cloud after one scan is rather challenging. With the real experience from the difficulties in data generation for 3D dairy cow reconstruction such as the movement of objects, problems of data collection with one camera and hardware limitations when recording, we built a system that combines 3D reconstruction and point cloud registration with the aim of creating a full dairy cow 3D point cloud.

The system is divided into two main parts. One is data collection that uses the stereo data series as input to generate the RGB-D image data. In this part, the depth image was generated based on the state-of-the-art CREStereo convolutional neural network and compared with other depth images generated by traditional methods. Part two is the dairy cow reconstruction initialization system. The RGB-D dataset in step one used as input in this step for creating a fragment point cloud of a dairy cow. They will be registered through the point cloud registration algorithm and refined again to create the final full 3D point cloud of the dairy cow. Particularly, the contribution of this paper can be summarized as follows:We improved the depth image quality based on a convolutional neural network through the stereo data inputs;The 3D reconstruction framework is proposed to increase the accuracy of 3D point cloud registration from objects with little motion;Generated point clouds could be used as the input data for dairy cow characteristics analysis with a deep learning approach.

The remainder of this paper is organized as follows: Section 2 describes the point cloud registration algorithm and their applications in 3D reconstruction tasks. In Section 3, a reconstruction framework for improving the dairy cow 3D point cloud quality is presented. Section 4 presents the experimental results and the evaluations of the proposed approach in the multiple dataset. Section 5 concludes this paper.

## 2. Related Work

Iterative closest point [2] is known as a widely used algorithm to determine alignment between two roughly aligned 3D point clouds. This algorithm searches for correspondence between two given point cloud sets and then optimizes object functions with the aim of minimizing the distance between corresponding points. However, ICP as well as some local refinement algorithms require a rough initial alignment as input. Normally, we can obtain the initial alignment using global registration algorithms [3,4].

In this work, the point-to-plane ICP [5] algorithm is applied due to its fast convergence based on the input set of gray images and depth images. In addition, the color-ICP algorithm [6] can be used for the purpose of increasing the accuracy of the surface alignment step if the input is color image data. Thus, the accuracy of the final reconstruction is also increased.

Many approaches to 3D reconstruction from RGB-D sequences have been explored [7,8,9]. These systems usually have three main steps: surface alignment (odometry and loop closure), global optimization, and surface extraction. According to the kinect fusion system [10], the real-time reconstruction with a depth camera was built on range image integration [11], visual odometry [12], and real-time 3D reconstruction [13]. This system does not detect loop closure, so it requires complicated camera routes for the comprehensive image of complex scenes.

Several RGB-D reconstruction systems with integrated loop closure have been developed [14,15]. The loop closure is detected by matching individual RGB-D images using visual features key points algorithms (SIFT, SURF) or through dense image registration. Real-time performance is improved with this approach, but it can miss loop closures that are not certified with the right image.

Our approach for milk cow reconstruction has similar framework as in [7]. We focused on dense surface reconstruction from RGB-D dataset and presented a dedicated method that defines outliers by optimizing the surface alignment directly. Instead of using traditional algorithms to generate depth images such as semi-global matching [16], we use a state-of-the-art method that is based on convolutional neural networks to improve the quality of depth images from stereo input data. The evaluation of the quality of the depth image between the methods is presented in detail in the experimental results, Section 4.

## 3. The Dairy Cow 3D Reconstruction Framework

In this paper, we proposed a 3D reconstruction framework that uses CNN for input data generation and applied SLAC algorithm that aim to improve accuracy in the dairy cow 3D point cloud registration problem. As indicated in Figure 1, the 3D reconstruction framework contains two main parts:Data recording based on the CNN;Dairy cow 3D reconstruction.

**Figure 1 sensors-22-09325-f001:**
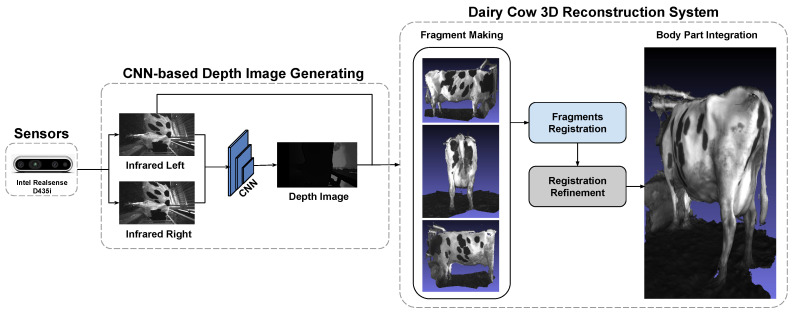
The dairy cow 3D reconstruction framework.

### 3.1. Data Preparing for Dairy Cow 3D Reconstruction

To overcome the drawbacks in the experimental procedure, such as hardware limitations and difficulties in scanning moving objects, we built a 3D point cloud scanning system for dairy cows through stereo images. In this approach, the RGB-D data are used as input to create a 3D point cloud of the dairy cow’s body parts. These parts then generate its entire body. Stereo matching is still a challenging problem and it has been studied for a long time. With the development of convolutional neural networks and the support of large synthetic data sets [17], we use learning-based algorithms for creating depth images from stereo data in the current work instead of using the traditional approach [16,18,19].

In this work, the pre-trained model based on the approach in [20] for creating depth image from stereo data was applied. As described in Figure 2, the depth map generating system was built based on the convolutional neural network. Specifically, the network is implemented with Pytorch framework. The model is trained on 8 NVIDIA GTX 2080Ti GPUs, with a batch size of 16. The whole training process is set to 300.000 iterations. The Adam optimizer was used with the learning rate of 0.0004. The learning rate is linearly increased from 5 to 100 of the standard value at the beginning of the training. After 180.000 iterations, the learning rate is linearly decreased down to 5 of the standard value to the end of the process. The model is trained with an input size of 384 × 512, all training samples undergo a set of augumentation operations before being fed into the model. The feature extractor network is described in Figure 3a. Moreover, the three-level feature pyramid will be generated through a shared-weight feature extraction network from a pair of stereo images, shown in Figure 3b. It is used to compute different scales of correlations in the three stages of a cascaded recurrent network. Additionally, the feature of the infrared left image also provides context information for update blocks and offsets computation. The features and the predicted disparities are refined using the Recurrent Update Module (RUM) in each stage. The final output disparity of the latter state is used for the next step as an initialization. For each iteration in RUM, the adaptive group correlation layer (AGCL) is applied for the correlation computation.

After creating the final disparity, the depth image is generated and combined with the infrared left image for the preparation of the RGB-D dataset through Equation (Equation 1).
(1)$$depth\_map=\frac{Bf}{disparity(valid\_pixel)}$$
where 

*B*: the distance between two cameras;

*f*: the focal length of camera;

valid_pixel=disparity>0.

### 3.2. Milk Cow 3D Reconstruction

The RGB-D data sequence generated above will be fed into the milk cow 3D reconstruction system as input. In our approach, the system will be divided into four specific steps as shown in Figure 1.

#### 3.2.1. Fragment Construction

First, we generated k-frame segments from short RGB-D sequences that already exist. For each sub-sequence, the RGB-D odometry [21] algorithm is used to estimate the camera trajectory and fuse the image range. Specifically, for adjacent RGB-D frames, the identity matrix is used as the initialization. In contrast, for non-adjacent RGB-D frames, the ORB feature is computed to match sparse features over wide baseline images [22], then performs 5-point RANSAC [23] for a rough alignment estimation that is used as the initialization of RGB-D odometry computation. Depending on the number of input frames to select the appropriate *k* value, *k* = 100 is set for all experiments with the current data. From the first 100 frames, we can create fragments that describe a part of the dairy cow surface mesh.

Suppose that an image (I) and a depth image (D) are registered to the same coordinate frame. With a pair of RGB-D images $\left(I_{i},D_{i}\right)\;\mathrm{and}\;\left(I_{j},D_{j}\right)$ and an initial transformation $T^{0}$, that roughly aligns $\left(I_{j},D_{j}\right)$ to $\left(I_{i},D_{i}\right)$. The current identified problem is how to find the optimal transformation that densely aligns the two given RGB-D images. As described in [6], we can build the optimization objective to obtain a tight alignment. Here, it is formatted by combining the objectives of photometric $E_{I}$ and geometric $E_{D}$, as shown in Equation (Equation 2). We obtained more robust and accurate results through aligning both the tangent plane and the normal direction locking than by using only one of the two objective functions.
(2)$$E\left(T\right)=\left(1-\sigma \right)E_{I}\left(T\right)+\sigma \;E_{D}\left(T\right)$$

In [21], the photometric objective $E_{I}$ is computed through the squared differences of intensities.
(3)$$E_{I}\left(T\right)=\sum_{x}^{}{\left(I_{i}\left(x^{\prime }\right)-I_{j}\left(x\right)\right)}^{2}$$
where, 

$x={\left(u,v\right)}^{⊤}$: pixel in $\left(I_{j},D_{j}\right)$

$x^{\prime }={\left(u^{\prime },v^{\prime }\right)}^{⊤}$ the corresponding pixel in $\left(I_{i},D_{i}\right)$

The correspondence between two RGB-D images is built by the depth pixel to 3D point in the camera space of $\left(I_{j},D_{j}\right)$ conversion with *T* transformation, and then projected onto the image plane of $\left(I_{i},D_{i}\right)$.
(4)$$x^{\prime }=g_{uv}\left(s\left(h\left(x,D_{j}\left(x\right)\right),T\right)\right)$$

The conversion from a depth pixel to a 3D point in homogeneous coordinates *h* is defined in Equation (Equation 5):(5)$$h\left(u,v,d\right)={\left(\frac{(u-c_{x})d}{f_{x}},\frac{(v-c_{y})d}{f_{y}},d,1\right)}^{⊤}$$
where 

$f_{x},f_{y}$: the focal length of camera;

$c_{x},c_{y}$: the principal point of camera;

*s*: rigid transformation.

We can see that g is the inverse function of *h*, it maps a 3D point to a depth pixel as shown in Equation (Equation 6):(6)$$g\left(s_{x},s_{y},s_{z},1\right)={\left(\frac{s_{x}f_{x}}{s_{z}}+c_{x},\frac{s_{y}f_{y}}{s_{z}}+c_{y},s_{z}\right)}^{⊤}$$

It should be noted that the $E_{I}$ and $E_{D}$ must be defined on the same parameterize domain.

#### 3.2.2. Fragments Registration

After the fragments of the scene are created, they need to be aligned in a global space. Global registration is known as the algorithm that does not require an alignment for initialization. It usually computes and provides a less tight alignment and is used as initialization of the local methods as ICP. For pairs of near fragments, we determined the rough alignment by the aggregating RGB-D odometry obtained from the fragment construction step. Otherwise, the FPFH algorithm [24] is performed for global registration.

To avoid the odometry drift problem, each pair of fragments $\left(P_{i},P_{j}\right)$ is tested to find overlapping pairs by a geometric registration algorithm. If the fragments have enough overlap when aligned, an associated transformation $T_{ij}$ creates a candidate loop closure between fragments *i* and *j*. Currently, there are many approaches to solving the problem of global registration. In the current work, we used FGR [3] to initialize alignment. To generate the initial correspondence set K=(p,q), the FPFH algorithm was used. Specifically, $F\left(P\right)=\left\{F(p):p\in P\right\}$ and $F\left(Q\right)=\left\{F(q):q\in Q\right\}$, where F(p) is the feature computed for point p via FPFH and the same for F(q). The *T* rigid transformation for *Q* to *P* alignment can be established by optimizing the objective function Equation (Equation 7) so that distances between corresponding points are minimized.
(7)$$E\left(T\right)=\sum_{(p,q)\in K}^{}\rho \left(∥p-Tq∥\right)$$

Here, ρ() is a robust penalty function that is used for the validation and pruning performance automatically without imposing additional computational costs. The current 3D point cloud reconstruction problem requires aligning multiple surfaces to obtain the full body of a dairy cow. Several methods are suggested and implemented with the aim of solving the multiway registration problem [25,26]. However, they still have some limitations such as high computation costs or suboptimal alignment of pairwise registration. We followed the approach presented in [3]. Instead of optimizing separate pairwise alignments and then synchronizing the results, a global registration objective can be directly optimized over all surfaces. The performance of the FGR algorithm was specifically evaluated and compared with RANSAC, one of the typical algorithms for calculating the alignment initialization matrix in the experimental result section.

#### 3.2.3. Registration Refinement

For reasons of performance increasing, the global registration is only used for a heavily down-sampled point cloud, and the results based on this method are also not tight. For this reason, we used local registration to further refine the point cloud alignment. Currently, point-to-plane ICP [27] is a commonly used algorithm in the local registration problem, and it was shown in [2] that the algorithm has a faster convergence speed than the point-to-point ICP algorithm. It uses a different objective function—Equation (Equation 8)
(8)$$E\left(T\right)=\sum_{(p,q)\in K}^{}{\left(\left(p-Tq\right)n_{p}\right)}^{2}$$
with $n_{p}$ as the normal of point.

Moreover, the authors of [6] presented the colored point cloud alignment algorithm that provides more accurate fragment alignment. Specifically, after dairy cow fragments are generated with initialized tight registrations, we ran the ICP iterations with a joint optimization objective as shown in Equation (Equation 9):(9)$$E\left(T\right)=\left(1-\delta \right)E_{c}\left(T\right)+\delta E_{G}\left(T\right)$$
where: 

*T*: the estimated transformation matrix;

$E_{C}$: the photometric component;

$E_{G}$: the geometric component;

δ∈[0,1]: weight parameter. 

If the photometric component is not used, the colored point cloud alignment algorithm is equivalent to a point-to-plane ICP algorithm. The geometric component’s optimization objective is built from the correspondence set in the current iteration *K* and the normal of point $p,n_{p}$, as shown in Equation (Equation 10).
(10)$$E_{G}\left(T\right)=\sum_{(p,q)\in K}^{}{\left(\left(p-Tq\right)n_{p}\right)}^{2}$$

The difference between the color of point *q* (C(q)) and the color of its projection on the tangent plane of *p* is measured by the color component $E_{C}\left(T\right)$ in Equation (Equation 11). Furthermore, a multi-scale registration scheme is applied to further improve efficiency.
(11)$$E_{C}\left(T\right)=\sum_{(p,q)\in \kappa }^{}{\left(C_{p}\left(f\left(Tq\right)\right)-C\left(q\right)\right)}^{2}$$

In the above equation, 

$C_{p}\left(x\right)$: precomputed function continuously defined on the tangent plane of *p*;

f(x): the function used to projects a 3D point to the tangent plane. 

For a better evaluation, a table comparing the performance as well as the speed of convergence of the point-to-point ICP, point-to-plane ICP, and color-ICP approaches is presented in Section 4.

#### 3.2.4. Dairy Cow Full Body Integration

For the RGB dataset, the depth image is synchronized and aligned. In the final step of milk cow 3D reconstruction, we integrated all RGB-D images into a single TSDF volume and then extracted a mesh of the dairy cow. Specifically, the alignment results from the fragment construction Section 3.2.1 and the fragment registration Section 3.2.2 step is used for the pose of each RGB-D image computation in the global space. Utilizing the RGB-D image integration algorithm [10], a dairy cow 3D mesh is reconstructed from the whole RGB-D sequence.

Currently, our main problem is dairy cow 3D point cloud reconstruction, so dairy cows moving more or less during data scanning is inevitable. As shown in [28], the simultaneous localization and calibration algorithms were applied to improve the final 3D point cloud of dairy cow quality, as well as to remove the redundant point clouds created by the movement of the dairy cow.

## 4. Experimental Results

### 4.1. Milk Cow Recording System

Our dairy cow scanning system consists of two main parts: hardware and software. The hardware includes a single-board computer and a depth camera. The hardware is built as shown in Figure 4. Specifically, the grip frame is fixed with two cameras located at the top and bottom view positions. The use of additional cameras placed in the top position in our system is intended to provide additional data that are missing during bottom camera data scanning because of field-of-view limitations. With this approach, the full 3D point cloud of dairy cow results will be reproduced more fully and the loss of the point cloud area leading to inaccurate measurement is also improved. The single-board computer is also fixed with a compact wireless source to facilitate data recording. The device specifications are detailed in Table 1.

Software: we built specialized software to collect the data read from the sensor through the communication between the tablet and the single board computer. It is designed based on the problem of analyzing dairy cow characteristics through its parts. Specifically, the software will extract stereo data with the unique ID of the cow, which we can use to extract its origin to facilitate the tracking and analysis process. As shown in the screen of the dairy cow data collection application, we can select the desired region to create 3D data from stereo data. In addition, our application also displays stereo images obtained from two cameras placed at the top and bottom positions. The stereo data will be uploaded and saved at server as input for the 3D reconstruction problem. The application interface is detailed in Figure 5.

### 4.2. Depth Image Evaluation

After obtaining stereo data, we proceeded to evaluate the depth image quality with the current approach. As shown in Figure 6, the object is clearly displayed with CNN-based depth image generation and the details are outlined by the bounding box in Figure 6b. Especially in the breast region of dairy cows, a very important part of the dairy characterization process is shown clearly and compared to the depth image of the other two approaches.

In addition, the black area phenomenon appears quite a lot in the overexposed areas or the border between the dairy cow and the background, as shown in Figure 6c,d. This results in 3D dairy cow information being lost during point cloud initialization.

For further verification, Figure 7 shows the generated point clouds corresponding to the depth images above. Dairy cow parts are visualized in detail via a 3D point cloud with our approach. Specifically in Figure 7b, the breast part can be created in detail. In contrast, point cloud information is missing in some parts of the dairy cow, which leads to errors in size estimation and analysis. The part of the lost point cloud information is marked with a red rectangle in Figure 7c,d.

### 4.3. Dairy Cow Point Cloud Registration Evaluation

After determining the method to create the best RGB-D dataset, we created a 3D point cloud for the dairy cow. First of all, we need to determine the influence of each part of the dairy cow’s body on reproductive quality, milk quality, and lifespan. As described in the theoretical part, to be able to create a 3D point cloud of each part of the dairy cow, the 3D point cloud registration approach was used. We focus on creating a 3D point cloud of each part first, then create a 3D point cloud for the whole cow body.

In our dairy cow reconstruction system, two cameras are calibrated for synchronizing the data acquisition process. To be able to create a complete dairy cow 3D point cloud, we record about 500 stereo frames with fps=15 in each camera. These data are used for the creation of the depth image, which is combined with the left image to create a 3D point cloud. After creating the body 3D point cloud fragment from 500 pairs of gray and depth images with $n_{fragment}=5$, we used the point cloud global registration algorithm for alignment for initialization. With the current approach of using the FGR algorithm, the accuracy of the algorithm is evaluated and compared with RANSAC, one of the most widely used algorithms in the point cloud global registration problem. As shown in column 1 of Table 2, our current approach has shown that the register two point cloud’s tightness is better and the processing time is also faster than the method, while remaining low. After determining the optimal algorithm for initialization of alignment, we performed the registration refinement so that the point cloud alignment between the target and the source point cloud would be tighter and more accurate. Specifically, we compared the current approach (local registration algorithm point-to-plane ICP) with two other algorithms, point-to-point ICP and color ICP. Since the alignment initialization in the previous step was good, there is not much change in the accuracy index of the register point cloud in this step. Specifically, the fitness-average value has not changed, but the RMSE-average value has decreased from 0.0063 to 0.0060.

To make our current results more visible, Figure 8 illustrates the point cloud registration process between two sets of point clouds that describe a dairy cow body part. Specifically, from the source point cloud in Figure 8a (generated from first 100 RGB-D frame) and the target point cloud in Figure 8b (generated from the next 100 RGB-D frames), we found the transform matrix so that the target point is fine-tuned to the closest match to the source point cloud. In Figure 8c is a description of two sets of non-registered point clouds (gray color represents source point cloud and blue color represents target point cloud), and then the result of two point clouds after the registration algorithm is applied in Figure 8d,e.

### 4.4. Dairy Cow Point Cloud Registration Improvement

With the current practice of creating a 3D point cloud of dairy cows, it is inevitable that the object will move during the input data collection process. We optimized the pose of each RGB-D image based on the point cloud registration algorithm and applied an RGB-D integration algorithm to generate the dairy cow 3D point cloud.

However, the final 3D point cloud of dairy cows still has some outliers that affect the evaluation of dairy cow quality, as shown in Figure 9. In order to overcome the existing limitations, we utilized the SLAC algorithm for improving the final point cloud result. A nonlinear calibration model for the camera was estimated and used to correct the distortion in the data. Therefore, SLAC implementation in dairy cow reconstruction reduces drift for explicit loop closure detection and gives a qualitatively cleaner dairy cow reconstruction. As shown in Figure 10, unnecessary point clouds have been eliminated.

To further evaluate our approach, a comparison between current approach and RGB-D SLAM algorithm was implemented.

Specifically, we reconstructed dairy cows based on ORB-SLAM algorithm [29] which processes RGB-D inputs to estimate camera trajectory and build a 3D map of the object. The ORB-SLAM system is able to close loops, relocate, and reuse its 3D map in real time on standard CPUs. The dairy cow 3D point cloud based on this method is shown in Figure 11.

For objects which have little motion, such as dairy cows, the approach using the ORB-SLAM2 algorithm still has some limitations such as the appearance of many outliers in the back part due to the swaying of the cow’s tail. This makes it difficult to analyze dairy cow characteristics because some part of the point cloud is obscured or distorted.

Additionally, RTAB-Map [30] was applied for dairy cow 3D point cloud evaluation. This approach uses pairs of depth and RGB images to construct a point map. The graph is created, each node on the graph contain RGB-D images with a corresponding odometry pose. In order to find a loop closure when the graph is updated, RTAB-Map compares the new image with all previous one in the graph. When a loop closure is found, graph optimization is performed to correct the pose in the graph. A point cloud from the RGB-D image is generated for each node in the graph. This point cloud is transformed by using the pose in the node, and the full 3D point cloud map of the dairy cow is created. The detailed result is shown in Figure 12. Similar to ORB-SLAM2, the dairy cow 3D point cloud result has not been improved much by this method, the outlier still appeared frequently.

Finally, dairy cow reconstruction based on our approach has produced point cloud results with only a few outliers, solving the problem that persists in the RGB-D SLAM approach. Ths is detailed in Figure 13.

## 5. Conclusions

In this paper, we proposed the framework for dairy cow 3D reconstruction that uses state-of-the-art CNN architecture to generate and improve the input data quality and exploits the SLAC-based algorithm for increasing the accuracy of the point cloud registration. Experiments with dairy cow data show that our approach results in impressive 3D reconstruction with very few outliers compared to the RGB-D SLAM approach. After creating and improving the quality of the dairy cow 3D point cloud dataset, we proceed with training and extracting its body part point cloud feature based on deep learning network architecture for the point cloud. Once the 3D point cloud feature of dairy cows is identified, the measurement and analysis of dairy cow quality will be conducted automatically based on deep learning instead of manually as before. The accurate reproduction of 3D point clouds in animals has created the premise for the combination of research reports related to the influence of individual cow parts on their physiological quality with neural networks. The current approach for simultaneous localization and calibration of a stream of range images is applicable not only to the dairy cow 3D reconstruction problem, but also to a variety of other types of objects, such as creating 3D maps for robot-related problems, reconstructing other animals for analysis of characteristics of each species, and scanning 3D objects in daily life applications.

## Figures and Tables

**Figure 2 sensors-22-09325-f002:**
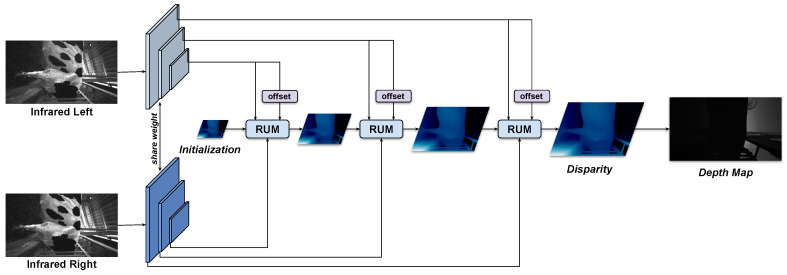
The depth map generating system using a learning-based algorithm.

**Figure 3 sensors-22-09325-f003:**
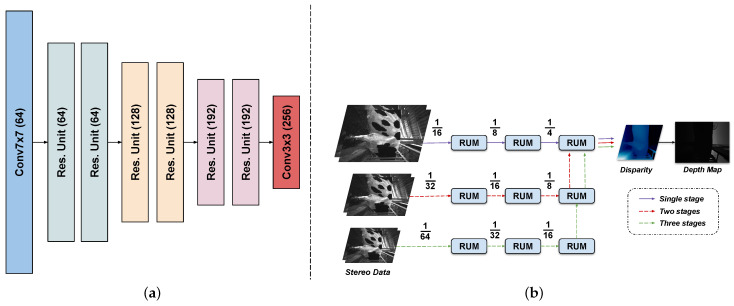
(**a**) The feature extractor network. (**b**) The stacked cascaded architecture in inference phase.

**Figure 4 sensors-22-09325-f004:**
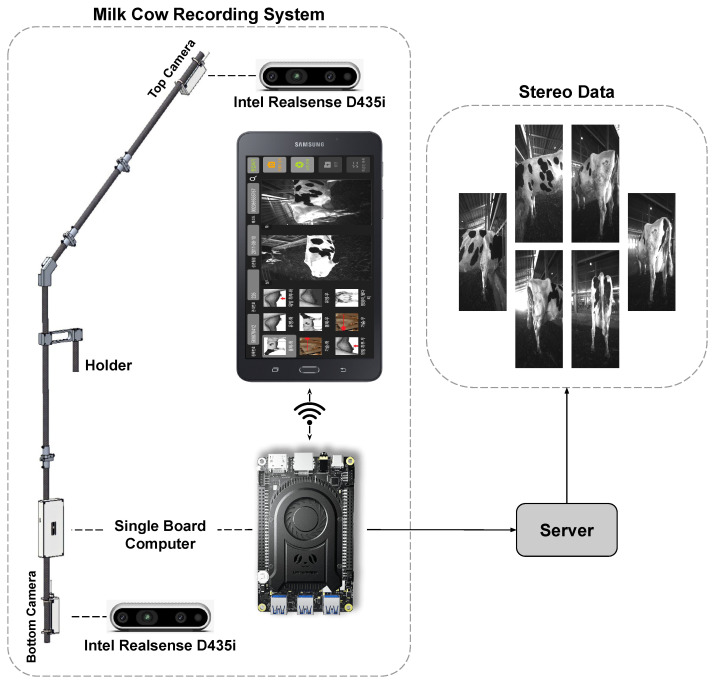
Dairy cow stereo dataset recording system.

**Figure 5 sensors-22-09325-f005:**
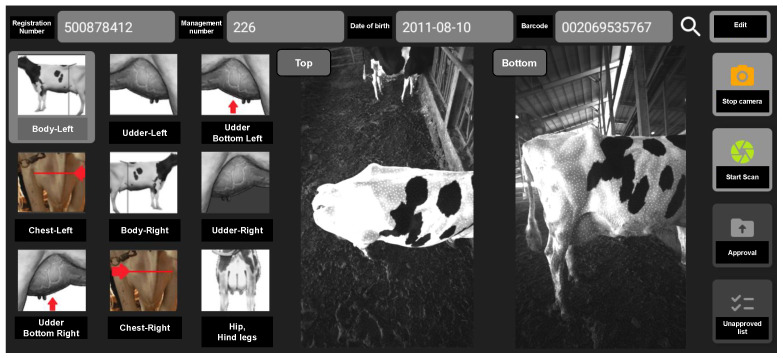
Dairy cow stereo dataset recording application.

**Figure 6 sensors-22-09325-f006:**
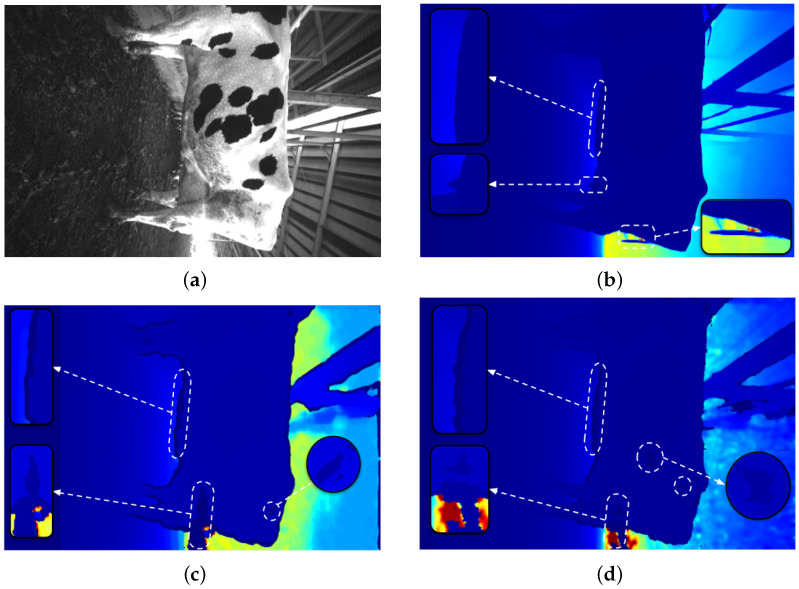
Depth image visual evaluation: (**a**) Gray image. (**b**) Our approach: depth mage generated based on CNN. (**c**) Depth image generated based on SGM algorithm. (**d**) Depth image from Intel realsense camera.

**Figure 7 sensors-22-09325-f007:**
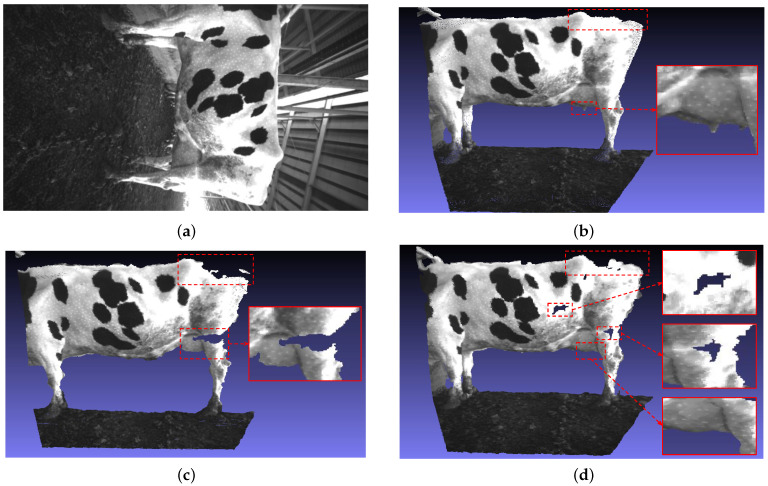
Point cloud visual evaluation: (**a**) Gray image. (**b**) Our approach: point cloud generated from CNN-based depth image generation. (**c**) Export point cloud from depth image that generated based on SGM algorithm. (**d**) Point cloud image Intel realsense camera.

**Figure 8 sensors-22-09325-f008:**
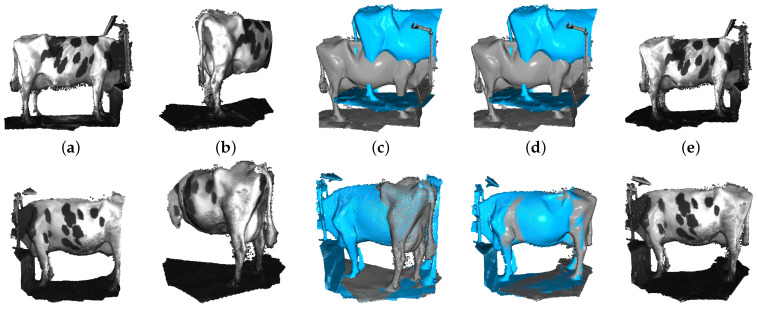
The process of point cloud registration: (**a**) Source point cloud. (**b**) Target point cloud. (**c**) Non-registered point cloud. (**d**) Our approach-based result: two point clouds after applying the registration algorithm. (**e**) Point cloud with dairy cow texture.

**Figure 9 sensors-22-09325-f009:**
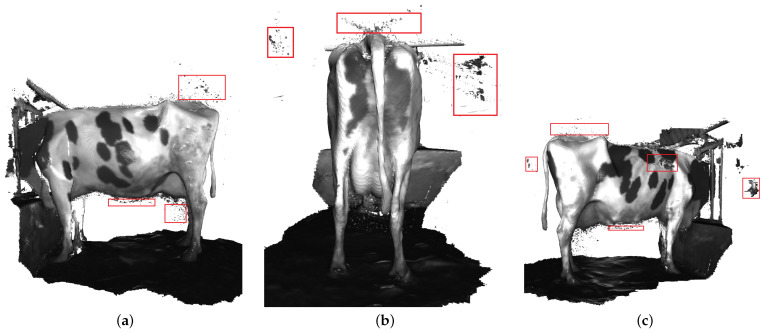
The dairy cow 3D reconstruction based on RGB-D Integration algorithm. Outliers are marked by the red bounding box. (**a**) Left side. (**b**) Back side. (**c**) Right side.

**Figure 10 sensors-22-09325-f010:**
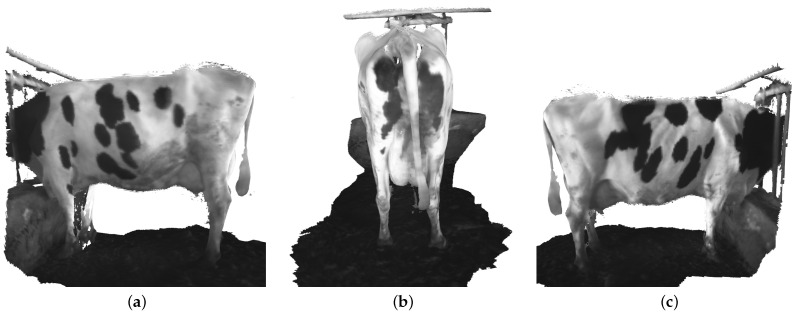
The dairy cow reconstruction after remove the outlier using SLAC algorithm. (**a**) Left side. (**b**) Back side. (**c**) Right side.

**Figure 11 sensors-22-09325-f011:**
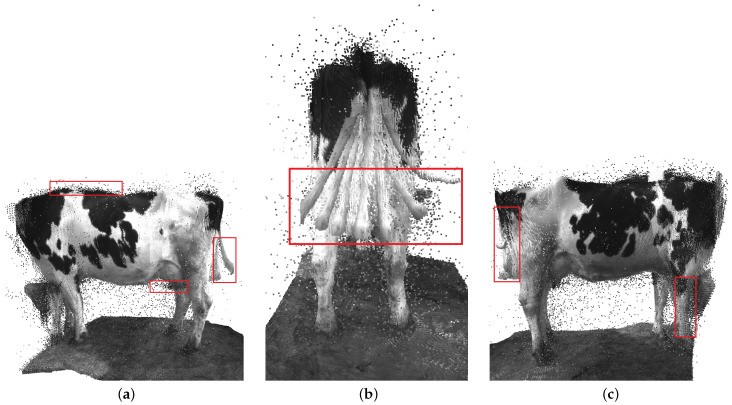
The dairy cow 3D reconstruction based on RGB-D SLAM, ORB-SLAM2. Outliers are marked by the red bounding box. (**a**) Left side. (**b**) Back side. (**c**) Right side.

**Figure 12 sensors-22-09325-f012:**
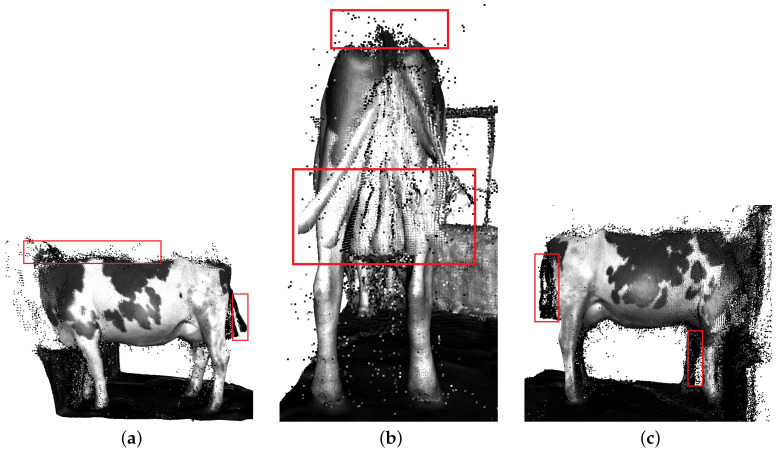
The dairy cow 3D reconstruction based on RGB-D SLAM, RTAB-map. Outliers are marked by the red bounding box. (**a**) Left side. (**b**) Back side. (**c**) Right side.

**Figure 13 sensors-22-09325-f013:**
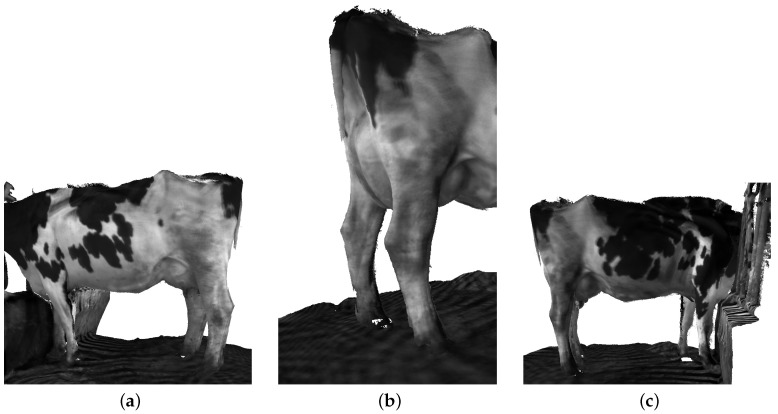
Our approach: dairy cow 3D reconstruction based on SLAC algorithm. (**a**) Left side. (**b**) Back side. (**c**) Right side.

**Table 1 sensors-22-09325-t001:** The specification of hardware device.

Device	Specification
Depth Camera (Intel Realsense D435i)	- Use environment: Indoor/Outdoor - Baseline [mm]: 50 - Resolution: 1920 × 1080 px - Frame rate: 30 fps - Sensor FOV (H × V × D): 69.4o × 42.5 × 77 (±3) - Dimensions: 90 × 25 × 25 mm - Connection: USB-C 3.1 Gen1
Single Board Computer (LattePanda 2 Alpha )	- Processor: Intel Core i5-8210Y - CPU Spec: 2-Core, 4-Thread, 1.60∼3.60 GHz - Memory: 8 Gb LDPPR3 1600 Hz - Storage: 60 Gb eMMC - Wireless: 802.11ac, 2.4G & 5G, up to 433 Mbps - Operating System: Window 10/Linux

**Table 2 sensors-22-09325-t002:** Dairy cow 3D Point Cloud Registration Evaluation.

Evaluation	Point Cloud Registration		Registration Refinement		
	*RANSAC*	*Current Approach*	*Point-to-Point ICP*	*Color ICP*	*Current Approach*
**Fitness-Average**	0.48	0.50	0.5	0.49	0.5
**RMSE-Average**	0.0065	0.0063	0.0061	0.0063	0.0060
**Time (s)**	14.86	12.99	1.59	2.88	2.15

**Fitness**: the overlapping area metrics, the higher the better. **Inlier RMSE**: the RMSE of all inlier correspondences metrics, the lower the better. **Time (s)**: the time of convergence.

## Data Availability

Not applicable.

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
