# Peer review of "Case Study: Improving the Quality of Dairy Cow Reconstruction with a Deep Learning-Based Framework"

_sensors, 2022, doi:10.3390/s22239325_

Round 1

Reviewer 1 Report

The work presented by the authors seems to be appropriate, the application is defined specifically which should reach to a conclusive point. 

1. In line 250&251, it is stated that a specialized software is built to collect data, if you can further improve in this context this will be informative with respect to readers.

2. The results are specified that, using two cameras the results are improved as per the statement in the abstract but it is not clear about what extent of quality is improved.

3. If the drawbacks of previous method are addressed with current approach, this needs to justified with scientific judgement with comparative results.

4. The conclusion needs to be further improved with context to the defined problem in support of work.

Reviewer 2 Report

Dear authors,

Thank you for your contribution! The usage of artificial intelligence for domestic and industrial applications is gaining a reasonable popularity nowadays. There are some suggestions to improve your manuscript:

1) Please, provide a sufficient literature research with recent references. Current references do not describe the latest situation.

2) The template must be followed.

3) The quality of Pictures could be improved.

4) Please, format Tables and Equations.

5) Conclusion must be completed. Please, make a proper summary about conducted research. What would be the future work?

Reviewer 3 Report

1. In the 3D reconstruction phase, the author only implemented one method. I recommend more comparisons of different methods in this phase.

2. What is the neural structure of CNN? What is the size of the filter? How many neurons and layers were used?

3. Did the author train the model by themself, or is it a pre-trained model? If yes, what is the loss function and optimizer? What if the training hardware and how long did it take? The setting and the experiment of the training process should be fully discussed.

4. How much data did the author collect?

5. What is the FPS of the proposed method?

Reviewer 4 Report

The paper proposes a construction of image through 3D point and some improvement was made. The problem statement was clear. The research methodology was clearly described. The setup and result were clearly discussed. The language was good but some minor grammar checking is needed.

The major part of deep learning part is missing. The authors need to explain this part clearly in the paper.

Round 2

Reviewer 3 Report

The author has replied to all my concerns.